# A Cross-Cultural Study of Social Support from Family, Friends, and Significant Others and Mental Health Among Italian, Spanish, and Portuguese Gay and Lesbian Young Adults

**DOI:** 10.3390/ijerph22071038

**Published:** 2025-06-29

**Authors:** Nicola Picone, Pedro Alexandre Costa, Marta Evelia Aparicio-García, Gaetana Affuso

**Affiliations:** 1Department of Psychology, University of Campania “Luigi Vanvitelli”, 81100 Caserta, Italy; nicola.picone@unicampania.it; 2Center for Psychology of University of Porto, Faculty of Psychology and Educational Sciences, University of Porto, 4200-135 Porto, Portugal; pacosta@fpce.up.pt; 3Department of Social, Work and Differential Psychology, Universidad Complutense de Madrid, 28223 Madrid, Spain; meaparic@ucm.es

**Keywords:** social stigma, social support, mental health, gay and lesbian young adults, cross-cultural study

## Abstract

Gay and lesbian young adults report worse mental health than their heterosexual counterparts due to social stigma. Nonetheless, factors such as social support may protect them against the negative effects of stigma. The current study compares Italian, Spanish, and Portuguese gay and lesbian young adults on three dimensions of social support (family, friends, and significant others) and on mental health indicators (depression and anxiety). It also explores the associations between social support and mental health among the three countries. To this end, a sample of 687 gay and lesbian young adults filled out an online questionnaire: 345 from Italy (*M*_age_ = 25.24, *SD* = 4.39), 193 from Spain (*M*_age_ = 27.44, *SD* = 5.05), and 149 from Portugal (*M*_age_ = 24.45, *SD* = 5.15). Italian participants reported lower levels of social support from family and friends than Spanish participants. Moreover, Portuguese participants reported higher levels of anxiety than Italian participants. The results of the survey further indicate that only social support from family was significantly and negatively associated with depression and anxiety in the three countries. Overall, the findings suggest that the mental health of gay and lesbian young adults can be improved through non-stigmatising cultures as well as family education.

## 1. Introduction

### 1.1. Mental Health and Social Support Among Gay and Lesbian People: The Role of Context

Gay and lesbian people present worse mental health outcomes than heterosexual people because of social stigma [1,2,3]. Specifically, according to the minority stress model (MSM) [4], sexual minoritised people are exposed to distal stressors (experiences of discrimination and violence) and proximal stressors (identity concealment, internalised stigma, expectations of rejection) that negatively affect their mental health. For instance, Cochran et al. [5] found that, compared with heterosexual people, gay and bisexual men reported higher levels of panic disorder, depression and psychological distress, while lesbian and bisexual women exhibited higher levels of generalised anxiety disorder. The same study also found that sexual minoritised people were more likely to use mental health services. As concerns behavioural outcomes, Kuhlemeier et al. [6] showed that lesbian, gay, bisexual, and questioning youth were more at risk of suicidal behaviour and substance use than heterosexual youth. Furthermore, Powdthavee and Wooden [7] observed that lesbian, gay, and bisexual (LGB) people reported low satisfaction with their lives, which was explained by variables such as having children, income, and education, among others.

Nonetheless, Meyer [8] argues that some variables may work as protective factors against the negative effects of minority stress. Specifically, the author highlights that sexual minoritised people may rely on personal characteristics (e.g., hardiness) or so-called ‘minority coping’ (e.g., social support provided by communities) to counteract the social stigma that they endure. Perrin et al. [9] also posit a minority strengths model in which several dimensions (e.g., self-esteem, identity pride, etc.) are thought to have a positive effect on sexual and gender minoritised people’s mental health and behaviours. Several authors have underlined the protective role that social support may play for the mental health of gay and lesbian people. However, the sources of social support, including family, friends, and significant others [10], may vary among sexual minoritised people. For example, a study revealed that family is the main source of support for lesbian and bisexual women, whereas friends fill this role for gay and bisexual men [11]. Importantly, Watson et al. [12] found that parental support was associated with lower levels of depression and higher levels of self-esteem among sexual minoritised youth. Relatedly, another study carried out by McConnell et al. [13] showed that lesbian, gay, bisexual, and transgender (LGBT) youth who received low family support were at risk of psychological distress. Furthermore, Detrie and Lease [14] demonstrated that support from both family and friends had a positive association with LGB youth’s psychological well-being. Lastly, Calvo et al. [15] identified a negative association between perceived social support (from family, friends, and significant others) and internalised stigma in a sample of gay men.

Given that stigma derives from social context [1], its effects on mental health and social support among gay and lesbian people may differ according to the country of residence considered. Following the ecological framework [16], the macrosystem, which includes laws, norms, and belief systems, can influence individuals and their relationships with others. For instance, prevailing religious systems impact gay and lesbian people’s experiences and mental health both negatively and positively [17]. They increase distal stressors (e.g., rejection by the family) and proximal stressors (e.g., internalised stigma and identity conflict) because heteronormativity is inherent to many religions [17]. Conversely, they also offer ways to cope with stressors, as evidenced by research with lesbian Muslims who use their religious beliefs as a coping strategy, for instance, by attending online support groups [18]. Similarly, keeping in mind the concept of macrosystem, gay and lesbian people may experience diverse levels of mental health and social support depending on the laws, policies, and institutional protection of their context of residence. In this regard, Pachankis et al. [19] highlighted that sexual minoritised men living in highly stigmatising countries (in terms of laws and social attitudes towards homosexuality) suffered from depression and suicidality more than those who live in less stigmatising countries. Specifically, the authors found that structural stigma was indirectly associated with depression and suicidality through social isolation, internalised stigma, and the concealment of sexual orientation (only for suicidality). That is, gay men from highly stigmatising contexts were more likely to receive low social support, experience internalised stigma and conceal their identity, which led to worse mental health. Another study revealed that sexual minoritised people’s life satisfaction depends on the country of residence, with lower levels observed in highly stigmatising contexts [20]. Lastly, research has found that illicit substance use is more common among sexual minoritised individuals living in states with high structural stigma (e.g., those that do not recognise same-sex marriage or have fewer gay–straight alliances in schools) [21].

### 1.2. The Present Study

In light of this, one of the aims of the present study is to analyse the impact of the national context on the mental health and social support of Italian, Spanish, and Portuguese gay and lesbian young adults. Italy, Spain, and Portugal share similarities in terms of language, religion, and historical background. They are Romance-language countries where a high percentage of the population identifies as Catholic. Moreover, they have all gone through a fascist regime that left no space for civil rights. Nowadays, the three countries differ greatly in their laws and social policies for lesbian, gay, bisexual, transgender, queer/questioning, and other gender and sexually diverse (LGBTQ+) people’s rights. According to the latest ranking published by ILGA-Europe in 2024 [22], Italy is considered the most stigmatising country of the three. It has yet to introduce same-sex marriage and adoption and lacks an anti-discrimination law. On the contrary, Spain and Portugal have already recognised these rights for a long time. For example, same-sex marriage and adoption were legalised in 2005 in Spain and, in 2010 and 2016, respectively, in Portugal. Nonetheless, Portugal has encountered some hardships on the journey towards LGBTQ+ rights [23].

In light of the literature examining the role of social support, another aim of this study is to analyse the direct associations between three dimensions of social support (i.e., family, friends, and significant others) and depression and anxiety.

Specifically, we hypothesise that:

**H1.** 
*Given their highly stigmatising climate, Italian participants should present lower levels of the three dimensions of social support and higher levels of anxiety and depression than Spanish and Portuguese participants.*


**H2.** 
*Given the literature on the role of social support (e.g., McConnell et al. [13]), the three dimensions of social support should be negatively associated with anxiety and depression in Italy, Spain, and Portugal.*


Because previous research showed the effects of age [24], sexual orientation [25], type of living area [26], level of education [27], and parents’ level of education [28], we incorporated these variables into the analyses as covariates.

To the best of our knowledge, this is the first study to analyse these factors from a cross-cultural perspective focusing on Italian, Spanish, and Portuguese gay and lesbian young adults. Its results will inform future interventions in line with Objective 10 of the 2030 United Nations Agenda [29] regarding inclusivity and the reduction of inequalities.

## 2. Materials and Methods

### 2.1. Participants

The final sample comprised 687 gay and lesbian young adults: 345 from Italy (*M*_age_ = 25.24, *SD* = 4.39), 193 from Spain (*M*_age_ = 27.44, *SD* = 5.05), and 149 from Portugal (*M*_age_ = 24.45, *SD* = 5.15). In Italy, 62.6% of the participants identified as gay and 37.4% as lesbian; in Spain, 74.6% were gay and 25.4% lesbian; in Portugal, 45.5% were gay and 55.5% lesbian. Furthermore, 69.9% of Italian participants, 69.1% of Spanish participants, and 78.8% of Portuguese ones lived in urban areas. Most participants had a high level of education (holding at least a university degree) in the three countries: 55.4% in Italy, 81.9% in Spain, and 61.7% in Portugal. Participants who had mothers with a high level of education represented 21.2% of the sample in Italy, 38.5% in Spain, and 34% in Portugal. Participants with highly educated fathers accounted for 20.6% in Italy, 31.9% in Spain, and 26.4% in Portugal. Sociodemographic information by nationality is displayed in Table 1.

### 2.2. Procedure

The participants filled out an online questionnaire (in their respective language) comprised of measures already used in other studies. The questionnaire was disseminated in the three countries through a non-probabilistic sampling method. The same dissemination channels were used in the three countries. Specifically, the study was distributed through the research team’s contacts (e.g., social activists), LGBTQ+ associations, universities (e.g., offices dedicated to inclusion and diversity), social media (e.g., Instagram) and at Pride events. People were asked to participate and/or forward the invitation to other people who might be eligible to take part in the study. Participants were asked to consent before taking part in the study. The same eligibility criteria were used in the three countries: participants were required to identify as a lesbian or gay person, be 18–35 years old, be Italian, Spanish, or Portuguese and currently live in their country of nationality. Before starting the survey, participants were asked to answer a question about their eligibility for the study. They were not rewarded for their participation. Filling out the questionnaire took approximately 15–20 min.

The study was approved by the Research Ethical Board of the Department of Psychology of the University of Campania ‘Luigi Vanvitelli’. The American Psychological Association’s ethical standards regarding research with human subjects were followed throughout the research design and implementation.

### 2.3. Measures

#### 2.3.1. Sociodemographic Information

We collected information about age, sexual orientation (1 = gay, 2 = lesbian), type of living area (1 = urban, 2 = suburban/rural), level of education, and participants’ mothers’ and fathers’ levels of education. In the analyses, mothers’ and fathers’ levels of education were combined into a single latent factor.

#### 2.3.2. Perceived Social Support from Family, Friends, and Significant Others

The Multidimensional Scale of Perceived Social Support was employed to measure support from family, friends, and significant others [10]. The Italian-, Spanish-, and Portuguese-language versions were already used in other studies [30,31,32,33,34,35]. This scale comprises three dimensions, each containing four items and measuring the social support provided by family (“I get the emotional help and support I need from my family”), friends (“My friends really try to help me”), and significant others (“There is a special person who is around when I am in need”). Participants were asked to report their agreement with each item on a Likert-type scale of 1 (very strongly disagree) to 7 (very strongly agree). In the three dimensions, higher scores represent greater perceived social support. For the family dimension, Cronbach’s alphas were 0.93, 0.96, and 0.94 for Italian, Spanish, and Portuguese participants, respectively. For the friends dimension, Cronbach’s alphas were 0.94, 0.95, and 0.97 for the Italian, Spanish, and Portuguese samples, respectively. Finally, for the significant other dimension, Cronbach’s alphas were 0.94 for Italian participants, 0.89 for Spanish ones, and 0.93 for Portuguese ones.

#### 2.3.3. Depression and Anxiety

To measure depression and anxiety, we used the short form of the Patient-Reported Outcomes Measurement Information System for emotional distress and depression/anxiety [36,37,38]. The Italian-, Spanish-, and Portuguese-language versions were already employed in previous studies [39,40,41,42,43,44,45,46,47]. The depression scale comprises eight items, and the anxiety scale contains seven. Participants’ responses were rated on a Likert-type scale of 1 (never) to 5 (always) describing the frequency of depressive and anxiety symptoms experienced. Example items are “I felt sad” (depression) and “I felt tense” (anxiety). Higher scores represent higher levels of depression and anxiety. Cronbach’s alphas were 0.95 in Italy, 0.96 in Spain, and 0.95 in Portugal for the depression scale and 0.93 in Italy, 0.95 in Spain, and 0.92 in Portugal for the anxiety scale.

### 2.4. Data Analysis

Before carrying out the analyses, we assessed the assumptions of normality. Our study variables presented acceptable values for skewness and kurtosis [48]. Specifically, the skewness values ranged from −1.20 to 0.38, and the kurtosis values ranged from −0.95 to 0.88. Next, we performed a multivariate analysis of covariance (MANCOVA) to explore the impact of the national context on the three dimensions of social support (family, friends, and significant other), depression, and anxiety. Therefore, we included nationality as the independent variable and social support (family, friends, and significant other), depression, and anxiety as dependent variables. Furthermore, we incorporated age, sexual orientation, type of living area, level of education, and parents’ level of education as covariates. For the significant associations between the covariates and the dependent variables, we examined the *t*-statistic associated with each regression coefficient.

To test the associations between social support and depression and anxiety (H2), we used multigroup structural equation modelling. Specifically, we tested a model in which the three dimensions of social support were predictors, while anxiety and depression were outcomes. In this model, we treated age, sexual orientation, type of living area, level of education, and parents’ level of education as covariates influencing all variables. First, we tested a model in which the parameters were estimated freely in the three countries. Next, we ran a model in which the parameters were constrained to be the same in the three countries. Maximum likelihood estimations were used [49]. To evaluate our model, we considered the comparative fit index (CFI) [50], the Tucker–Lewis index (TLI) [51], and the root mean square error of approximation (RMSEA) [52]. The model was accepted if the indexes presented the following values: CFI and TLI ≥ 0.90 and RMSEA ≤ 0.08 [53]. Moreover, the Satorra–Bentler chi-square difference test (ΔSBχ^2^) was performed to compare the fit of nested models [54]. We used the Statistical Package for the Social Sciences (version 25) and Mplus 7.4 to conduct the analyses.

## 3. Results

### 3.1. Descriptive Statistics and Correlations

The MANCOVA indicated a significant impact of the national context: Wilks’s λ = 0.93, *F* (10, 1324) = 4.60, *p* < 0.001, η^2^ = 0.03. Specifically, the post-hoc univariate analysis revealed significant differences between the three countries regarding social support from family, social support from friends, and anxiety. The means and standard deviations are displayed in Table 2. The Bonferroni post-hoc comparisons showed that Italian participants had lower levels of social support from both family and friends than Spanish participants. Moreover, Portuguese participants had higher levels of anxiety than Italian participants.

The parameter estimates demonstrated significant associations of the covariates with the study variables. Notably, sexual orientation (1 = gay, 2 = lesbian) was significantly and negatively associated with social support from friends (*t* = −0.25; *p* < 0.05) and significantly and positively associated with social support from a significant other (*t* = 0.32; *p* < 0.05) and anxiety (*t* = 0.33; *p* < 0.001). The type of living area (1 = urban, 2 = suburban/rural) presented a significant and negative association with social support from family (*t* = −0.33; *p* < 0.05) and from a significant other (*t* = −0.31; *p* < 0.05), while it had a significant and positive association with depression (*t* = 0.21; *p* < 0.05). The level of education was significantly and positively associated with social support from friends (*t* = 0.26; *p* < 0.05) and significantly and negatively associated with both depression (*t* = −0.38; *p* < 0.001) and anxiety (*t* = −0.18; *p* < 0.05). Lastly, the parents’ level of education had a significant and positive association with social support from family (*t* = 0.21; *p* < 0.01) and a significant and negative association with anxiety (*t* = −0.10; *p* < 0.05).

The Pearson correlations of the study variables are displayed in Table 3, Table 4 and Table 5 for Italy, Spain, and Portugal, respectively.

### 3.2. Associations Between the Three Dimensions of Social Support and Depression and Anxiety

We tested the first model, which had the following fit indexes: χ^2^(26) = 26.82, *p* = 0.42, RMSEA = 0.01 (0.00; 0.05), TLI = 1.00, and CFI = 1.00. Next, we constrained the parameters to be the same for the three countries, and the model produced the following fit indexes: χ^2^(116) = 173.68, *p* = 0.00, RMSEA = 0.05 (0.03; 0.06), TLI = 0.96, and CFI = 0.96. The delta chi-square statistic indicated that the fit of the model was worse: Δχ^2^(90) = 146.86, *p* < 0.001. Therefore, examining each constraint, we improved the fit by estimating the correlations between age and level of education and parents’ level of education, and the correlation between the type of living area and level of education freely in the three countries. The new model presented a good fit to the data: χ^2^(110) = 130.28, *p* > 0.05, RMSEA = 0.03 (0.00; 0.05), TLI = 0.98, and CFI = 0.99. The delta chi-square statistic was non-significant (Δχ^2^(84) = 103.46, *p* > 0.05), indicating that this model could be accepted. The model is displayed in Figure 1 for Italy, Spain, and Portugal. In the three countries, social support from family presented a significant and negative association with both anxiety and depression, while the other two dimensions of social support did not show any significant associations.

The associations between the covariates and the other variables are reported in Table 6 and Table 7. In the three countries, sexual orientation (1 = gay, 2 = lesbian) had a significant and positive association with both social support from a significant other and anxiety. That is, being lesbian was associated with higher levels of social support from a significant other and anxiety. In addition, we observed a significant and negative association between the type of living area (1 = urban, 2 = suburban/rural) and two dimensions of social support (family and significant other), where suburban/rural participants received less support from family and significant others than urban ones. Moreover, the level of education was significantly and positively associated with social support from friends, but significantly and negatively associated with both depression and anxiety. Finally, the parents’ level of education had a significant and positive association with social support from family.

The variance explained was 20% for depression and 12% for anxiety in Italy, 19% for depression and 9% for anxiety in Spain, and 18% for depression and 13% for anxiety in Portugal.

## 4. Discussion

The present study explored the impact of the national context on mental health and social support among Italian, Spanish, and Portuguese gay and lesbian young adults. It also examined the associations between three dimensions of social support and depression and anxiety in the three countries.

We found that Italian participants presented lower levels of two dimensions of social support (family and friends) than Spanish participants. This finding may be explained by Italian culture, which is more stigmatising against gay and lesbian people. According to the ecological framework [16], macrosystems including laws, norms, and religious beliefs can have an impact on people’s mental health and social support. As a reflection of its stigmatising culture, Italy has yet to recognise rights such as same-sex marriage and adoption and approve an anti-discrimination law. As previous research suggests [19], gay and lesbian young adults residing in highly stigmatising countries may suffer more and be more isolated than those living in progressive countries. Working towards a non-stigmatising culture may decrease the detrimental effects of social stigma and increase sexual minoritised people’s levels of perceived social support. Nevertheless, although Portugal is considered a more progressive country than Italy [22], our results revealed that Portuguese participants reported higher levels of anxiety than Italian participants, which may be due to reasons other than culture. Further investigation is therefore needed.

Regarding the associations of the three dimensions of social support (family, friends, and significant others) with depression and anxiety, we found that in the three countries, controlling for the associations of the covariates, only social support from family was significantly and negatively associated with the two mental health outcomes, while the other two dimensions did not present any significant associations. This finding echoes the literature about the protective role of family support for the mental health of sexual minoritised people [12,13]. Although the bivariate correlations showed that the other two dimensions of social support (friends and significant others) also correlated with anxiety and depression, these associations disappeared when they were included in the structural equation modelling. Overall, our findings suggest that family may play a more important role than other agents as a social determinant of lesbian and gay young adults’ mental health across different contexts (Italy, Spain, and Portugal). The literature about family reactions to coming out supports the assumption that family has a specific impact on sexual minoritised people’s experiences, with negative parental reactions linking to higher levels of internalised stigma [55] and depression [56].

As concerns the other variables, we found that sexual orientation (1 = gay, 2 = lesbian) was significantly and positively associated with both social support from a significant other and anxiety. That is, lesbian participants received more social support from their significant other and were more likely to suffer from anxiety than gay participants, which may be accounted for by the intersection of sexuality and gender [57]. Male homosexuality is usually less accepted than female homosexuality because of the inflexibility of male gender roles across contexts [58]. As a result, gay men may receive less social support because they do not fulfil said norms. At the same time, the higher levels of anxiety observed among lesbian participants may be explained by stressors and difficulties related to the patriarchal structures embedded in our societies [59]. Women overall report worse mental health than men [60].

The type of living area (1 = urban, 2 = suburban/rural) was also significantly and negatively associated with two dimensions of social support (family and significant others). That is, suburban/rural participants received less support from their families and significant others. This suggests that social stigma may be even greater in suburban or rural areas, impacting the levels of perceived social support. This finding is in line with past research showing that LGB youth living in rural areas face greater social isolation [26].

Furthermore, the level of education had a significant and positive association with social support from friends as well as significant and negative associations with both depression and anxiety. This suggests that education is a protective factor: well-educated young adults may be better able to seek out social support and cope with depression and anxiety. The protective role of education in mental health has also been observed in previous studies [27,61]. Similarly, our results showed that the parents’ level of education was significantly and positively associated with social support from family. Parents with higher levels of education may be more prone to show support to their gay and lesbian children, which connects to the literature about the predictive role of education for attitudes towards homosexuality [62].

The present study has some limitations. First, we used a non-probabilistic sampling method and a cross-sectional design, which limited the generalisability of our results and the conclusions about causal relationships among the study variables. Second, the sample size differed in the three countries. Third, we did not include all subgroups in the LGBTQ+ community in our sample but only gay and lesbian participants. We decided not to include other identities (e.g., bisexual people) because prior research pointed to substantial differences between subgroups of the LGBTQ+ community. For example, Chan et al. [63] showed that bisexual people report higher levels of depression and anxiety than gay and lesbian individuals. Moreover, they face specific struggles and discrimination within the LGBTQ+ community itself [64]. Consequently, we decided to include only gay and lesbian people so as not to overcomplicate the cross-cultural comparison between the three countries. However, we are aware that including other identities would have made the study richer. Moreover, because of our sample size, we were not able to stratify our results by sexual orientation. Such analyses may offer important clinical implications.

Previous research has explored the topic in a cross-cultural perspective [65]. However, to the best of our knowledge, ours is the first study to explore these factors in Italian, Spanish, and Portuguese gay and lesbian young adults. In line with Objective 10 of the 2030 United Nations Agenda [29], our results can be used to inform interventions aimed at reducing inequalities and promoting gay and lesbian people’s mental health.

## 5. Directions for Future Research

Future cross-cultural research may overcome the limitations of our study. First, longitudinal research is necessary to clarify the direction of the relations that we found among our study variables. For example, the relations may also be reversed, such that mental health indicators (depression and anxiety) influence people’s perception of social support. Second, future cross-cultural research using probabilistic sampling methods can overcome limitations to the generalisability of our results. Moreover, larger sample sizes may help researchers stratify their analyses by sexual orientation to reveal important implications. Fourth, recruiting participants with other identities within the LGBTQ+ community (e.g., bisexual people) may allow researchers to highlight similarities and differences among subgroups from a comparative perspective considering countries/contexts. For example, experiences of social support and mental health may be the same for homosexual and bisexual people living in one country but different for those living in another. Lastly, including other variables (e.g., connectedness to the LGBTQ+ community) from a cross-cultural perspective may allow researchers to reveal which dimensions remain protective regardless of the country/context.

## 6. Conclusions and Implications

Our results shed light on the role of stigmatising macrosystems (including laws, norms, and religious beliefs) in influencing people’s mental health and protective factors. We found that Italian participants reported lower levels of two dimensions of social support (family and friends) than Spanish participants, which may be due to their respective contexts. Thus, working towards a non-stigmatising culture may increase social support and improve mental health among gay and lesbian young adults. Nonetheless, in line with the ecological framework [16], interventions should be implemented in other systems as well (e.g., family settings). Importantly, we found that social support from family was the only dimension with a significant and negative association with depression and anxiety in all three countries, which suggests that family support may be a protective factor for gay and lesbian young adults’ mental health. Based on our results, interventions should be designed keeping in mind the intersection of sexuality and gender, and they should be implemented above all in suburban and rural areas where levels of social support may be lower. Overall, interventions should primarily target families (particularly those with lower levels of education) to educate them about sexuality and the challenges and stressors that gay and lesbian young adults endure because of social stigma [1]. This may boost the support offered to sexual minoritised young adults by their families, which is associated with better mental health.

## Figures and Tables

**Figure 1 ijerph-22-01038-f001:**
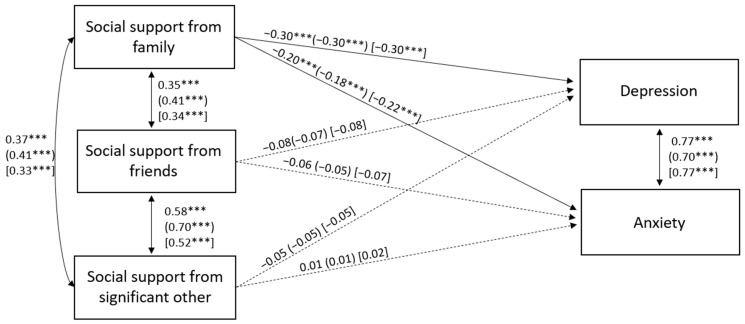
Associations among social support from family, social support from friends, social support from significant other, depression, and anxiety by nationality. Standardized path coefficients for Italy (outside brackets), Spain (in round brackets), and Portugal (in square brackets) are displayed. Solid and dotted lines represent significant and non-significant paths, respectively. To simplify, the associations of the covariates with the other variables are not depicted, but they are reported in Table 6 and Table 7. Note: *** *p* < 0.001.

**Table 1 ijerph-22-01038-t001:** Sociodemographic characteristics by nationality.

	Italian Young Adults	Spanish Young Adults	Portuguese Young Adults
Characteristics	*n* (%) or *M* (*SD*)	*n* (%) or *M* (*SD*)	*n* (%) or *M* (*SD*)
Age	25.24 (4.39)	27.44 (5.05)	24.45 (5.15)
Sexual orientation			
Lesbian	129 (37.4%)	49 (25.4%)	82 (55%)
Gay	216 (62.6%)	144 (74.6%)	67 (45%)
Type of living area			
Urban	241 (69.9%)	152 (78.8%)	103 (69.1%)
Suburban/Rural	104 (30.1%)	41 (21.2%)	46 (30.9%)
Level of education			
High	191 (55.4%)	158 (81.9%)	92 (61.7%)
Low	154 (44.6%)	35 (18.1%)	57 (38.3%)
Mothers’ level of education			
High	73 (21.2%)	74 (38.5%)	50 (34%)
Low	272 (78.9%)	118 (61.5%)	97 (66.1%)
Fathers’ level of education			
High	71 (20.6%)	61 (31.9%)	39 (26.4%)
Low	273 (79.4%)	130 (68.1%)	109 (73.7%)

**Table 2 ijerph-22-01038-t002:** Means and standard deviations for social support from family, social support from friends, social support from significant other, depression, and anxiety by nationality.

	Italian Young Adults		Spanish Young Adults		Portuguese Young Adults		
Variables	*Mean*	*SD*	*Mean*	*SD*	*Mean*	*SD*	*F* (2, 673)
Social support from family	4.14 ^a^	1.72	4.97 ^b^	1.79	4.34 ^ab^	1.72	9.89 ***
Social support from friends	5.56 ^a^	1.46	5.99 ^b^	1.35	5.69 ^ab^	1.39	3.69 *
Social support from significant other	5.66	1.60	5.70	1.52	5.66	1.61	0.11
Depression	2.60	1.12	2.48	1.08	2.71	1.06	0.45
Anxiety	2.72 ^a^	1.04	2.71 ^ab^	1.09	3.10 ^b^	0.90	4.69 *

Note: Bonferroni post-hoc comparisons: a < b; *** *p* < 0.001, * *p* < 0.05.

**Table 3 ijerph-22-01038-t003:** Correlations among the study variables in Italy.

	A	SO	TLA	LE	PLE	SSF	SSFR	SSO	DEP	AN
A	-									
SO	−0.23 ***	-								
TLA	−0.05	−0.03	-							
LE	0.49 ***	−0.18 **	−0.23 ***	-						
PLE	0.01	0.14 **	−0.08	0.09	-					
SSF	0.11 *	−0.11 *	−0.14 **	0.09	0.18 **	-				
SSFR	0.01	−0.15 **	−0.08	0.09	0.02	0.29 ***	-			
SSO	−0.02	0.05	−0.07	0.02	−0.03	0.33 ***	0.58 ***	-		
DEP	−0.10	0.13 *	0.16 **	−0.24 ***	−0.12 *	−0.35 ***	−0.23 ***	−0.19 **	-	
AN	−0.14 **	0.18 **	0.09	−0.20 ***	−0.13 *	−0.21 ***	−0.18 **	−0.08	0.80 ***	-

Note: *** *p* < 0.001, ** *p* < 0.01, * *p* < 0.05. A = Age, SO = Sexual Orientation, TLA = Type of Living Area, LE = Level of Education, PLE = Parents’ Level of Education, SSF = Social Support from Family, SSFR = Social Support from Friends, SSO = Social Support from Significant Other, DEP = Depression, AN = Anxiety.

**Table 4 ijerph-22-01038-t004:** Correlations among the study variables in Spain.

	A	SO	TLA	LE	PLE	SSF	SSFR	SSO	DEP	AN
A	-									
SO	−0.16 *	-								
TLA	−0.03	0.02	-							
LE	0.02	−0.02	−0.14	-						
PLE	−0.27 ***	0.11	−0.16 *	0.05	-					
SSF	−0.02	−0.03	−0.12	0.23 **	0.05	-				
SSFR	−0.10	0.05	−0.14 *	0.22 **	0.12	0.55 ***	-			
SSO	0.04	0.10	−0.13	0.24 **	0.00	0.52 ***	0.72 ***	-		
DEP	−0.03	−0.01	0.07	−0.24 ***	−0.01	−0.40 ***	−0.27 ***	−0.34 ***	-	
AN	−0.06	0.10	0.00	−0.10	−0.07	−0.27 ***	−0.07	−0.10	0.71 ***	-

Note: *** *p* < 0.001, ** *p* < 0.01, * *p* < 0.05. A = Age, SO = Sexual Orientation, TLA = Type of Living Area, LE = Level of Education, PLE = Parents’ Level of Education, SSF = Social Support from Family, SSFR = Social Support from Friends, SSO = Social Support from Significant Other, DEP = Depression, AN = Anxiety.

**Table 5 ijerph-22-01038-t005:** Correlations among the study variables in Portugal.

	A	SO	TLA	LE	PLE	SSF	SSFR	SSO	DEP	AN
A	-									
SO	−0.11	-								
TLA	−0.21 *	0.05	-							
LE	0.25 **	−0.11	0.06	-						
PLE	−0.04	0.06	−0.23 **	−0.05	-					
SSF	0.14	−0.00	−0.05	0.03	0.13	-				
SSFR	0.11	−0.06	−0.01	0.03	−0.02	0.33 ***	-			
SSO	0.13	0.15	−0.11	−0.13	−0.02	0.21 *	0.40 ***	-		
DEP	−0.12	0.09	0.08	−0.14	−0.12	−0.44 ***	−0.25 **	−0.08	-	
AN	−0.17 *	0.20 *	0.14	−0.11	−0.04	−0.36 ***	−0.17 *	−0.04	0.77 ***	-

Note: *** *p* < 0.001, ** *p* < 0.01, * *p* < 0.05. A = Age, SO = Sexual Orientation, TLA = Type of Living Area, LE = Level of Education, PLE = Parents’ Level of Education, SSF = Social Support from Family, SSFR = Social Support from Friends, SSO = Social Support from Significant Other, DEP = Depression, AN = Anxiety.

**Table 6 ijerph-22-01038-t006:** Associations of age, sexual orientation, type of living area, level of education, and parents’ level of education with social support from family, social support from friends, and social support from significant other by nationality.

Covariates	Social Support from Family	Social Support from Friends	Social Support from Significant Other
Age	0.04, 0.05, 0.05	−0.05, −0.06, −0.05	0.02, 0.03, 0.02
Sexual orientation	−0.07, −0.07, −0.07	−0.07, −0.07, −0.07	0.10 *, 0.10 *, 0.09 *
Type of living area	−0.09 *, −0.08 *, −0.08 *	−0.07, −0.07, −0.06	−0.09 *, −0.08 *, −0.08 *
Level of education	0.08, 0.08, 0.07	0.10 *, 0.11 **, 0.09 *	0.06, 0.07, 0.05
Parents’ level of education	0.12 *, 0.13 *, 0.12 *	0.05, 0.06, 0.05	−0.05, −0.05, −0.04

Note: ** *p* < 0.01, * *p* < 0.05. Standardized path coefficients are displayed for Italy, Spain, and Portugal, respectively. Significant correlations between Sexual Orientation and Parents’ Level of Education (*β* = 0.13, *p* < 0.01; *β* = 0.14, *p* < 0.01; *β* = 0.13, *p* < 0.01, respectively, for Italy, Spain, and Portugal), Age (*β* = −0.18, *p* < 0.001; *β* = −0.17, *p* < 0.001; *β* = −0.15, *p* < 0.001, respectively, for Italy, Spain, and Portugal), Level of Education (*β* = −0.11, *p* < 0.01; *β* = −0.13, *p* < 0.01; *β* = −0.12, *p* < 0.01, respectively, for Italy, Spain, and Portugal); between Type of Living Area and Parents’ Level of Education (*β* = −0.13, *p* < 0.01; *β* = −0.15, *p* < 0.01; *β* = −0.14, *p* < 0.01, respectively, for Italy, Spain, and Portugal), Level of Education (*β* = −0.24, *p* < 0.001; only for Italy); between Age and Level of Education (*β* = 0.49, *p* < 0.001; *β* = 0.27, *p* < 0.01; only for Italy and Portugal), Parents’ Level of Education (*β* = −0.30, *p* < 0.001; only for Spain).

**Table 7 ijerph-22-01038-t007:** Associations of age, sexual orientation, type of living area, level of education, and parents’ level of education with depression and anxiety by nationality.

Covariates	Depression	Anxiety
Age	−0.00, −0.00, −0.00	−0.06, −0.07, −0.08
Sexual orientation	0.05, 0.05, 0.05	0.13 ***, 0.12 **, 0.15 ***
Type of living area	0.05, 0.05, 0.05	0.03, 0.02, 0.03
Level of education	−0.16 ***, −0.16 ***, −0.15 ***	−0.09 *, −0.08 *, −0.09 *
Parents’ level of education	−0.04, −0.04, −0.04	−0.08, −0.07, −0.08

Note: *** *p* < 0.001, ** *p* < 0.01, * *p* < 0.05. Standardized path coefficients are displayed for Italy, Spain, and Portugal, respectively.

## Data Availability

Data are available from the corresponding author on reasonable request.

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
