# Peer review of "A Cross-Cultural Study of Social Support from Family, Friends, and Significant Others and Mental Health Among Italian, Spanish, and Portuguese Gay and Lesbian Young Adults"

_ijerph, 2025, doi:10.3390/ijerph22071038_

Round 1

Reviewer 1 Report

Comments and Suggestions for Authors

Thank you very much for the opportunity to review this important article. I propose accepting the article with minor revisions. Firstly, as I am not a quantitative researcher, I will refrain from critiquing the analysis and focus on the theoretical framework and discussion. In my opinion, the theoretical framework at the beginning of the article should provide more information on different ways in which LGBTQ+ individuals cope with stigma and discrimination, particularly in contexts that historically marginalized them. For example, contemporary religious communities seeking to provide equal space for LGBTQ+ individuals to rectify years of injustice. Referencing important research by Dr. Elazar Ben-Lulu on Reform Jewish communities or DR. Rodriguez, Eric's studies on LGBTQ+ churches could be beneficial. Given that this article addresses a European country where religion plays a central role in shaping residents' identities, I suggest in the conclusion linking the micro to the macro and providing a broader explanation of how this case study may shed light on other communities and places. If in the introduction, the author of the article provides a broader understanding of how LGBTQ+ communities in different regions and cultures cope with discrimination and acceptance, it will be easier for him to expand the discussion, explain the findings on the data, and shed light on broader insights.

Author Response

Thank you very much for the opportunity to review this important article. I propose accepting the article with minor revisions. Firstly, as I am not a quantitative researcher, I will refrain from critiquing the analysis and focus on the theoretical framework and discussion. In my opinion, the theoretical framework at the beginning of the article should provide more information on different ways in which LGBTQ+ individuals cope with stigma and discrimination, particularly in contexts that historically marginalized them. For example, contemporary religious communities seeking to provide equal space for LGBTQ+ individuals to rectify years of injustice. Referencing important research by Dr. Elazar Ben-Lulu on Reform Jewish communities or DR. Rodriguez, Eric's studies on LGBTQ+ churches could be beneficial. Given that this article addresses a European country where religion plays a central role in shaping residents' identities, I suggest in the conclusion linking the micro to the macro and providing a broader explanation of how this case study may shed light on other communities and places. If in the introduction, the author of the article provides a broader understanding of how LGBTQ+ communities in different regions and cultures cope with discrimination and acceptance, it will be easier for him to expand the discussion, explain the findings on the data, and shed light on broader insights.

Thank you for your suggestion. We have broadened the introduction through the concept of macrosystem (according to the ecological framework by Bronfenbrenner) and research on religion (lines 99-107). Specifically, we have referenced research showing both the stressful and protective role that religious systems can play in the lives of gay and lesbian people. Then, we have pointed to that in our discussion (lines 442-443) and conclusions (lines 550-551) as well.

Reviewer 2 Report

Comments and Suggestions for Authors

Review of IJERPH-3687665 " Social support from family, friends and significant other and 2 mental health in Italian, Spanish, and Portuguese gay and les-3 bian youth: a cross-cultural study "

The present manuscript presented results of a cross-sectional survey study which was designed to examine differences in social support and mental health between lesbian women and gay men in Italy, Portugal, and Spain. Results show that

Italian participants reported the lowest levels of social support from family and friends. Portuguese participants reported the highest anxiety. In addition, the authors found that social support from family was significantly associated with depression and anxiety among participants. This manuscript sets out to explore how structural stigma across countries impacts mental health among members of the LGBT community. Below I’ve included my specific comments/critiques.

  1. There are minor typos/grammatical errors throughout this manuscript. Please revise.
  2. The Introduction section is structured oddly. The first five paragraphs give a brief overview of the literature then lay out the aims. Then this same information is repeated in more detail in sections 1.1-1.3. I would recommend using a more traditional academic article structure and remove the first five paragraphs.
  3. Lines 109-111. The authors could improve their argument by more thoroughly describing Pachankis et al.’s findings - that is the social isolation men experienced (lack of support) was due in part to structural stigma.
  4. The Procedures section could be expanded- I have several questions about hiw the study was conducted:
    1. Can the authors provide information on the recruitment procedures? It is unclear how similar the participant populations recruited were across countries. I am concerned that where participants were recruited may have impacted the findings. To me, this is potentially a significant problem with the manuscript- I have no way of knowing if these samples are comparable.
    2. Why was the sample restricted to only lesbian and gay identified participants? The authors note in the Discussion section this was a limitation- why were other identities (e.g., bisexual identified people) not recruited?
    3. Why was the sample restricted to those ages 18-35? The authors refer to the sample as “youth” in the Discussion, but this age range is not what I would consider “young”.
  5. I found the analyses somewhat confusing. Describing results such as “s, sexual orientation (1 = gay, 2 = lesbian) had a significant and positive association with both social support from significant other and anxiety” was hard to follow. If I am understanding the manuscript, the authors compared lesbian women to gay men and found that lesbian participants reported more social support from a significant other and higher anxiety. This could be made more explicit in the Results section (although this becomes clearer in the Discussion).
    1. Relatedly I feel like this comparison goes beyond what the authors set out to do. This suggests that structural stigma interacts with gender identity to impact behavior. Have the authors considered stratifying their results by gender? This could expand upon their analysis and offer important clinical implications.
  6. Frankly, I found the interpretation of the results to be a bit of a stretch. For instance, the statement “We found that Italian participants presented lower levels of two dimensions of social support (family and friends) than Spanish participants. This finding may be explained by the Italian social and political context which still has not introduced rights such as same-sex marriage and adoption nor has it approved an anti-discrimination law. …Although Portugal is a more progressive country than Italy [6], its political history was characterized by certain hardships on the journey toward LGBTQ+ rights [21]; which may be an explanation for our results about anxiety. For instance, Portugal legalized same-sex marriage in 2010, but at that time a new form of discrimination was introduced given that it was not until 2016 that adoption for same-sex couples was passed as well.” I think a more accurate interpretation is that the culture is more stigmatizing towards LGBT people. That is, the laws reflect the culture such that places where LGBT identities are still stigmatized are not going to pass laws protecting the community. Similarly, anxiety could be higher in Portugal for any of a myriad of reasons (e.g., when the data were collected and where participants were recruited from). I do believe that structural stigma impacts mental health, but I think in the current study the measures used are too blunt to accurately assess this.

Author Response

The present manuscript presented results of a cross-sectional survey study which was designed to examine differences in social support and mental health between lesbian women and gay men in Italy, Portugal, and Spain. Results show that

Italian participants reported the lowest levels of social support from family and friends. Portuguese participants reported the highest anxiety. In addition, the authors found that social support from family was significantly associated with depression and anxiety among participants. This manuscript sets out to explore how structural stigma across countries impacts mental health among members of the LGBT community. Below I’ve included my specific comments/critiques.

  1. There are minor typos/grammatical errors throughout this manuscript. Please revise.

Thank you for your suggestion. The manuscript has been revised by Cambridge Proofreading LLC Company to correct spelling mistakes and, more generally, to make the text more fluent and straightforward in English.

  1. The Introduction section is structured oddly. The first five paragraphs give a brief overview of the literature then lay out the aims. Then this same information is repeated in more detail in sections 1.1-1.3. I would recommend using a more traditional academic article structure and remove the first five paragraphs.

Thank you for your suggestion. We have changed the structure of our introduction.

  1. Lines 109-111. The authors could improve their argument by more thoroughly describing Pachankis et al.’s findings - that is the social isolation men experienced (lack of support) was due in part to structural stigma.

Thank you for your suggestion. We have described Pachankis et al.’s findings in more detail.

  1. The Procedures section could be expanded- I have several questions about hiw the study was conducted:
    1. Can the authors provide information on the recruitment procedures? It is unclear how similar the participant populations recruited were across countries. I am concerned that where participants were recruited may have impacted the findings. To me, this is potentially a significant problem with the manuscript- I have no way of knowing if these samples are comparable.

We have provided more information on the recruitment procedures (lines 218-229). As stated in the manuscript, we used the same recruitment procedures in the three countries. Moreover, we controlled potential differences among the three samples due to some sociodemographic characteristics (age, sexual orientation, type of living area, level of education, and parents’ level of education) given that we included such variables as covariates in our analyses.

    1. Why was the sample restricted to only lesbian and gay identified participants? The authors note in the Discussion section this was a limitation- why were other identities (e.g., bisexual identified people) not recruited?

We decided not to include other identities (e.g. bisexual people) because prior research pointed to substantial differences between subgroups of the LGBTQ+ community. For example, Chan et al. (2020) showed that bisexual people report higher levels of depression and anxiety than gay and lesbian individuals. Moreover, they face specific struggles and discrimination within the LGBTQ+ community itself (Velasco et al., 2024). Consequently, we decided to include only gay and lesbian people so as not to overcomplicate the cross-cultural comparison between the three countries. However, we are aware that including other identities would have made the study richer. Future research may overcome this limitation.

We have added this explanation to our limitations section (lines 513-521).

    1. Why was the sample restricted to those ages 18-35? The authors refer to the sample as “youth” in the Discussion, but this age range is not what I would consider “young”.

According to the APA dictionary (definition of “adulthood”) and other studies regarding the mental health of sexual minoritized people (Baiocco, Ioverno, Cerutti, Santamaria, Fontanesi, Lingiardi, Baumgartnert, & Laghi, 2014; Baiocco, Ioverno, Lonigro, Baumgartner, & Laghi, 2015), the age range 18-35 may be defined as young adulthood. Therefore, we have replaced “youth” with “young adults” accordingly.

  1. I found the analyses somewhat confusing. Describing results such as “s, sexual orientation (1 = gay, 2 = lesbian) had a significant and positive association with both social support from significant other and anxiety” was hard to follow. If I am understanding the manuscript, the authors compared lesbian women to gay men and found that lesbian participants reported more social support from a significant other and higher anxiety. This could be made more explicit in the Results section (although this becomes clearer in the Discussion).

Thank you for your suggestion. We have made it clearer in the Results section as well (lines 377-378).

    1. Relatedly I feel like this comparison goes beyond what the authors set out to do. This suggests that structural stigma interacts with gender identity to impact behavior. Have the authors considered stratifying their results by gender? This could expand upon their analysis and offer important clinical implications.

Thank you for your suggestion. We have considered stratifying our results, but the sample size did not allow us to do so (e.g., testing the associations between our study variables in the three countries only for gay participants vs testing them only for lesbian participants). Nevertheless, we included sexual orientation as a covariate influencing all other variables in our analyses.

We are aware that this could offer important clinical implications, thus we added that as a limitation of our study (lines 521-523)

  1. Frankly, I found the interpretation of the results to be a bit of a stretch. For instance, the statement “We found that Italian participants presented lower levels of two dimensions of social support (family and friends) than Spanish participants. This finding may be explained by the Italian social and political context which still has not introduced rights such as same-sex marriage and adoption nor has it approved an anti-discrimination law. …Although Portugal is a more progressive country than Italy [6], its political history was characterized by certain hardships on the journey toward LGBTQ+ rights [21]; which may be an explanation for our results about anxiety. For instance, Portugal legalized same-sex marriage in 2010, but at that time a new form of discrimination was introduced given that it was not until 2016 that adoption for same-sex couples was passed as well.” I think a more accurate interpretation is that the culture is more stigmatizing towards LGBT people. That is, the laws reflect the culture such that places where LGBT identities are still stigmatized are not going to pass laws protecting the community. Similarly, anxiety could be higher in Portugal for any of a myriad of reasons (e.g., when the data were collected and where participants were recruited from). I do believe that structural stigma impacts mental health, but I think in the current study the measures used are too blunt to accurately assess this.

Thank you for your suggestion. We have changed our interpretation of the results through the concept of culture and macrosystem (according to the ecological framework by Bronfenbrenner). Then, we have changed the interpretation of the results regarding anxiety in Portugal (lines 440-455).

Reviewer 3 Report

Comments and Suggestions for Authors

I sincerely appreciate the opportunity to review the manuscript titled “Social support from family, friends, and significant other and mental health in Italian, Spanish, and Portuguese gay and lesbian youth: a cross-cultural study” submitted to the special issue Mental Health Challenges Affecting LGBTQ+ Individuals and Communities. It is always a privilege to contribute to the advancement of research in this field, and I am grateful for the trust placed in me for this evaluation.

Below, I provide my detailed comments, aimed at supporting the authors in refining their work and enhancing its clarity, methodological rigor, and overall contribution to the field.

Abstract:

1. I suggest avoiding starting a sentence with “Because,” as this can compromise clarity and readability in academic writing.

  1. In English, when listing three or more items, a comma should be placed before the final “and” (i.e., the Oxford comma). For example: “family, friends, and significant other.” Please also revise the manuscript title accordingly.
  2. Please consider rephrasing the sentence fragment: “and on two of mental health (depression and anxiety).” A more suitable version might be: “on mental health indicators (i.e., depression and anxiety).”
  3. As a native Spanish and Portuguese speaker, I notice that some syntactic structures seem influenced by Latin languages. I recommend that the manuscript be reviewed by a native English speaker or professional editor. For example, the sentence “Moreover, we explored the associations of social support with mental health in the three countries” would be more accurate as: “Moreover, we explored the associations between social support and mental health among the three countries.”
  4. Please avoid using the term well-being as a synonym for symptoms like depression and anxiety. Well-being is a broader construct that encompasses more than psychological symptoms. Use specific terms when referring to symptomatology.

Introduction:

  1. The introduction is somewhat difficult to follow. I recommend using clearer transitions between paragraphs and ideas to improve the overall flow.
  2. The English throughout this section should be reviewed for clarity, as some sentences are hard to parse.
  3. It would be helpful for readers if the authors justify the selection of Italy, Spain, and Portugal. Are there cultural, historical, or policy-related reasons for focusing on these countries?

Methods:

  1. It is unclear why participants identifying as plurisexual or with other minoritized sexual orientations were excluded. Please justify this decision.
  2. Including a comparative table with the sociodemographic characteristics of the samples across the three countries would enhance transparency and interpretability.
  3. Why was 35 chosen as the upper age limit? How do the authors conceptualize or define “youth” in this context?
  4. Please report all demographic variables collected (as outlined in Section 2.3.1), especially to facilitate future replication efforts.
  5. In the “Data Analysis” section, please indicate how the authors tested for assumptions of normality.

Results:

  1. Table 3 appears visually different from Tables 2 and 4. Please standardize formatting for consistency.
  2. Thank you for including Figure 1, it is helpful in visualizing key findings.
  3. Table 5 is difficult to follow in its current format. Consider restructuring it by organizing columns under each variable, or separating the sections for social support and mental health symptoms to enhance readability.

Discussion:

  1. The English writing in this section is much clearer, well done!
  2. Again, I encourage the authors not to equate mental health symptoms with well-being, as these are distinct constructs.
  3. The limitations section could be expanded. Some limitations, such as the exclusion of participants from other sexual orientations or the lack of sample comparability, could have been addressed during the development of the study design.
  4. I recommend including a subsection on directions for future research, highlighting how future studies might address the current study’s limitations or build on its findings.

Thank you again for the opportunity to review this manuscript. I hope these comments are helpful to the authors in revising their work.

Author Response

I sincerely appreciate the opportunity to review the manuscript titled “Social support from family, friends, and significant other and mental health in Italian, Spanish, and Portuguese gay and lesbian youth: a cross-cultural study” submitted to the special issue Mental Health Challenges Affecting LGBTQ+ Individuals and Communities. It is always a privilege to contribute to the advancement of research in this field, and I am grateful for the trust placed in me for this evaluation.

Below, I provide my detailed comments, aimed at supporting the authors in refining their work and enhancing its clarity, methodological rigor, and overall contribution to the field.

Abstract:

1. I suggest avoiding starting a sentence with “Because,” as this can compromise clarity and readability in academic writing.

Thank you for your suggestion. We have deleted it.

  1. In English, when listing three or more items, a comma should be placed before the final “and” (i.e., the Oxford comma). For example: “family, friends, and significant other.” Please also revise the manuscript title accordingly.

Thank you for your suggestion. We have revised it.

  1. Please consider rephrasing the sentence fragment: “and on two of mental health (depression and anxiety).” A more suitable version might be: “on mental health indicators (i.e., depression and anxiety).”

Thank you for your suggestion. We have rephrased it.

  1. As a native Spanish and Portuguese speaker, I notice that some syntactic structures seem influenced by Latin languages. I recommend that the manuscript be reviewed by a native English speaker or professional editor. For example, the sentence “Moreover, we explored the associations of social support with mental health in the three countries” would be more accurate as: “Moreover, we explored the associations between social support and mental health among the three countries.”

Thank you for your suggestion. The manuscript has been revised by Cambridge Proofreading LLC Company to correct spelling mistakes and, more generally, to make the text more fluent and straightforward in English.

  1. Please avoid using the term well-being as a synonym for symptoms like depression and anxiety. Well-being is a broader construct that encompasses more than psychological symptoms. Use specific terms when referring to symptomatology.

Thank you for your suggestion. We have revised the text accordingly.

Introduction:

  1. The introduction is somewhat difficult to follow. I recommend using clearer transitions between paragraphs and ideas to improve the overall flow.

Thank you for your suggestion. We have made changes to the introduction. Moreover, the manuscript has been revised by Cambridge Proofreading LLC Company to correct spelling mistakes and, more generally, to make the text more fluent and straightforward in English.

  1. The English throughout this section should be reviewed for clarity, as some sentences are hard to parse.

Thank you for your suggestion. As stated before, the manuscript has been revised by Cambridge Proofreading LLC Company.

  1. It would be helpful for readers if the authors justify the selection of Italy, Spain, and Portugal. Are there cultural, historical, or policy-related reasons for focusing on these countries?

Thank you for your suggestion. We have made our selection of the three countries clearer (lines 163-177)

Methods:

  1. It is unclear why participants identifying as plurisexual or with other minoritized sexual orientations were excluded. Please justify this decision.

We decided not to include other identities (e.g. bisexual people) because prior research pointed to substantial differences between subgroups of the LGBTQ+ community. For example, Chan et al. (2020) showed that bisexual people report higher levels of depression and anxiety than gay and lesbian individuals. Moreover, they face specific struggles and discrimination within the LGBTQ+ community itself (Velasco et al., 2024). Consequently, we decided to include only gay and lesbian people so as not to overcomplicate the cross-cultural comparison between the three countries. However, we are aware that including other identities would have made the study richer. Future research may overcome this limitation.

We have added this explanation to our limitations section (lines 513-521).

  1. Including a comparative table with the sociodemographic characteristics of the samples across the three countries would enhance transparency and interpretability.

Thank you for your suggestion. We have added a table with the sociodemographic characteristics of the samples across the three countries (Table 1).

  1. Why was 35 chosen as the upper age limit? How do the authors conceptualize or define “youth” in this context?

According to the APA dictionary (definition of “adulthood”) and other studies regarding the mental health of sexual minoritized people (Baiocco, Ioverno, Cerutti, Santamaria, Fontanesi, Lingiardi, Baumgartnert, & Laghi, 2014; Baiocco, Ioverno, Lonigro, Baumgartner, & Laghi, 2015), the age range 18-35 may be defined as young adulthood. Therefore, we have replaced “youth” with “young adults” accordingly.

  1. Please report all demographic variables collected (as outlined in Section 2.3.1), especially to facilitate future replication efforts.

Thank you for your suggestion. We have added a table with the sociodemographic characteristics of the samples across the three countries (Table 1).

  1. In the “Data Analysis” section, please indicate how the authors tested for assumptions of normality.

We have indicated that (lines 275-278).

Results:

  1. Table 3 appears visually different from Tables 2 and 4. Please standardize formatting for consistency.

We have done it.

  1. Thank you for including Figure 1, it is helpful in visualizing key findings.

  1. Table 5 is difficult to follow in its current format. Consider restructuring it by organizing columns under each variable, or separating the sections for social support and mental health symptoms to enhance readability.

Thank you for your suggestion. We have separated the sections for social support and mental health symptoms by adding another table (Table 6).

Discussion:

  1. The English writing in this section is much clearer, well done!

Thank you.

  1. Again, I encourage the authors not to equate mental health symptoms with well-being, as these are distinct constructs.

Thank you for your suggestion. We have revised the text accordingly.

  1. The limitations section could be expanded. Some limitations, such as the exclusion of participants from other sexual orientations or the lack of sample comparability, could have been addressed during the development of the study design.

Thank you for your suggestion. We have expanded our limitation section (lines 512-523).

  1. I recommend including a subsection on directions for future research, highlighting how future studies might address the current study’s limitations or build on its findings.

Thank you for your suggestion. We have added it.

Thank you again for the opportunity to review this manuscript. I hope these comments are helpful to the authors in revising their work.

Round 2

Reviewer 3 Report

Comments and Suggestions for Authors

I would like to thank you for the effort and rigour with which you have addressed the comments made in the previous round of review. The modifications introduced have significantly improved the clarity and relevance of the work.